# Circadian clock proteins KaiB and Rbp2 of *Synechococcus elongatus* display oscillations in their subcellular localization patterns

Harry J. Bevir,[1] Christopher C. Hooper,[1] Parker Saikley,[1] Tanya Chaljian,[1] Yahan Lin,[1] Susan E. Cohen[1,2]

**ABSTRACT** Circadian rhythms generated by molecular clocks time a diverse array of physiological cycles in many organisms. Cyanobacteria are currently the only prokaryotic system with a robust and rigorously tested circadian clock, which is carried out by KaiA, KaiB, and KaiC core oscillator proteins. This KaiABC oscillator drives global rhythms in gene expression, compaction of the chromosome, natural transformation, and timing of cell division in the model organism *Synechococcus elongatus* PCC 7942. The oscillator proteins have been previously shown to undergo changes in their subcellular localization patterns, where KaiA and KaiC are diffuse throughout the cell during the day and localize to or near one pole of the cell at night. However, functional fluorescent fusions of KaiB to track subcellular localization could not be obtained. More recently, we have described the identification of Rbp2 as a member of the extended clock network that associates with KaiC in a localized state. However, the subcellular localization patterns of KaiB and Rbp2 remain uncharacterized, and the mechanism driving the dynamic localization of circadian proteins remains undescribed. Here, we describe the patterns of KaiB and Rbp2 subcellular localization over the course of the day. Specifically, we demonstrate that KaiB and Rbp2 form polar foci and co-localize with KaiC at night using both traditional fusions to fluorescent proteins and endogenously expressed antibodies. We show that the RNA-binding activity of Rbp2 is necessary for the robust formation of Rbp2 foci. We propose a model by which Rbp2 drives circadian protein localization via its RNA binding activity.

**IMPORTANCE** Circadian rhythms driven by a circadian clock are required to time a wide range of physiological and metabolic processes. In cyanobacteria, the circadian clock is required for optimal fitness in environmental cycles. Where the clock proteins are located within the cell and how this helps them to synchronize with the environment and time rhythmic behaviors is incompletely understood. Here, we demonstrate that recently identified extended clock protein Rbp2 and core oscillator protein KaiB localize to the cell poles at night and co-localize with other members of the clock complex. We also demonstrate that the RNA binding activity of Rbp2 is necessary for the polar localization of Rbp2. Moreover, the use of endogenously expressed antibodies introduces a new and robust method for imaging challenging proteins in bacteria that is applicable across various fields.

**KEYWORDS** cyanobacteria, circadian, KaiB, Rbp2, KaiC

Address correspondence to Susan E. Cohen, scohen8@calstatela.edu.

The authors declare no conflict of interest.

See the funding table on p. 10.

Our planet experiences a consistent 24-h cycle of day and night as it rotates on its axis resulting in regular fluctuations in temperature, light intensity and color, and humidity. Organisms use these oscillating stimuli to entrain an internal molecular clock which times their physiological responses with daily environmental fluctuations,

known as circadian rhythms (1). These circadian rhythms enable organisms to optimize their biological timing by anticipating daily occurrences, providing a fitness advantage in cycling environments (2). Cyanobacteria, the planet's oldest oxygenic, photosynthetic organisms, are the only prokaryotes known to possess a robust and rigorously tested circadian clock. *Synechococcus elongatus* PCC 7942 (*S. elongatus*) is the premier model for studying the molecular details of the cyanobacterial circadian clock.

Cyanobacterial circadian rhythms are generated by a post-translational oscillator consisting of three central proteins: KaiA, KaiB, and KaiC (3). KaiC is a hexameric autokinase, autophosphatase, and ATPase (4, 5). Each KaiC monomer consists of a CI domain, a CII domain, and an A-loop tail. During the day, KaiA binds to the A-loops of unphosphorylated KaiC and promotes KaiC autophosphorylation (6). KaiC phosphorylates itself sequentially on Serine 431 and Threonine 432 until it is fully phosphorylated (4, 5). In its fully phosphorylated state, KaiC undergoes a conformational change that exposes a binding site for KaiB on the CI domain (7). KaiB mostly exists as a tetramer or dimer in a ground state fold conformation incapable of binding to KaiC. KaiB occasionally switches to a rare monomeric fold switched state with a thioredoxin like fold. In this state, it is capable of binding to KaiC (7). When bound to KaiC, KaiB binds and sequesters KaiA prompting the auto dephosphorylation of KaiC (8).

The oscillation of KaiC phosphorylation is synchronized with day/night cycles through redox sensing proteins such as KaiA (9) and CikA (10, 11) which bind to oxidized quinones in the thylakoid membranes. The clock is also synchronized by direct monitoring of ATP/ADP ratios through KaiC (12). The oxidation state of the quinone pool and ATP/ADP ratios all change as a function of day/night cycles allowing the clock to synchronize by monitoring the metabolic outputs of photosynthesis.

The KaiABC complex controls global rhythms of gene expression (13), chromosome compaction (14), natural transformation (15), and the timing of cell division (16). Circadian gene expression is controlled through the two-component output pathway consisting of SasA and RpaA (17). KaiC binds to SasA during the day, prompting SasA autophosphorylation and subsequent phosphotransfer to RpaA (17). Phosphorylated RpaA is a DNA binding transcription factor, which directly binds to ~100 targets in the genome, some of which are transcription factors themselves, resulting in a transcriptional cascade (18). At night, the association of CikA with the KaiABC complex promotes CikA's phosphatase activity where it removes the phosphate from RpaA (19). Through the cyclic activation of RpaA, the post translational oscillator impacts the expression patterns of most of the *S. elongatus* genome (18).

The subcellular localization of clock proteins has only recently been explored as a potential component of clock function. Through the use of functional fluorescent fusions to KaiA and KaiC, it was discovered that they are primarily diffuse during the day and co-localize to a focus at one pole of the cell at night, where KaiA polar localization is dependent on KaiC (20). This localization is circadian, with the number of cells containing polar foci increasing throughout the nighttime period and peaking at about 10 h after the onset of darkness, but only in the presence of a functional clock (20). Fluorescent fusions to CikA show that CikA co-localizes with KaiA and KaiC at night but remains localized to the pole at all circadian times (20, 21). The cyclic changes in subcellular organization could represent a mechanism that contributes to the robustness of the circadian clock.

To better understand the functionality and mechanisms behind the dynamic localization of clock components, we examined the localization of core oscillator protein KaiB and newly identified member of the extended clock network Rbp2. Rbp2 is an RNA binding protein that contains an RNA recognition motif (RRM) and was identified as associating with KaiC in a localized state (22). The deletion of *rbp2* results in long period rhythms of gene expression, in a manner that is dependent on Rbp2 ability to bind to RNA, and an approximately 50% reduction in KaiC polar localization at night (22). Taken together, these data suggest that RNA binding by Rbp2 is critical for its role in regulating the circadian clock and could be involved in circadian regulation of protein localization.

Here, we implement both traditional cell biological tools, i.e., fusions to fluorescent proteins, as well as the expression of endogenous antibodies to track the subcellular localization of proteins in *S. elongatus*. We observe that both KaiB and Rbp2 localize as a focus at or near the poles of cells and co-localize with KaiC, suggesting that both KaiB and Rbp2 are part of the nighttime polar complex. Additionally, we show that the use of endogenously expressed antibodies can be implemented to track proteins which cannot be observed using traditional cell biology methods in bacteria. We propose a model where RNA binding by Rbp2 is required for creating a scaffold that allows clock proteins to form complexes at or near the poles of cells which may facilitate protein complex assembly and association with the thylakoid membrane, suggesting that the subcellular localization of proteins would contribute to the robustness and the synchronization of the circadian clock in cyanobacteria.

## RESULTS

### Expression of endogenous antibodies or "frankenbodies" provides a method for imaging KaiB *in vivo*

KaiB is the only member of the core oscillator whose subcellular localization has not been determined as functional fluorescent fusion constructs could not be identified. KaiB is a small protein at only 11.4 KDa, and fluorescent proteins such as YFP are over twice as large at 27 KDa (23). As the KaiB function depends on an intricate fold switching mechanism (7), the addition of large fluorescent tags could affect KaiB dynamics, rendering fusions nonfunctional. To overcome this obstacle, we adapted a novel method for tracking the subcellular localization of KaiB *in vivo* using frankenbodies or endogenously produced antibodies (24). Frankenbodies are genetically encoded single-chain variable fragments that bind to either the HA or FLAG epitopes and labeled with a fluorophore, in this case Green Fluorescent Protein (GFP). They have been used to track protein dynamics in eukaryotic systems (24) (Fig. 1A).

In a strain with an unmarked deletion of the *kaiBC* operon (25), we expressed epitope tagged *kaiB* from Neutral Site (NS1), *kaiC* from NS2, and a luciferase reporter (P$_{kaiB}$-*luc*) in NS3 that allows us to monitor how epitope tagged variants of *kaiB* affect the circadian clock (7) (Fig. 1B). We found that strains expressing KaiB-FLAG, 1× FLAG tag on the C-terminus of KaiB, display wild-type (WT) rhythms of gene expression (Fig. S1A and B), suggesting that the FLAG tag does not interfere with KaiB function. In contrast, strains expressing KaiB-HA, 1×HA tag on the C-terminus of KaiB, displayed a short period phenotype (Fig. S1C and D), suggesting that the HA tag interferes with KaiB function. Strains lacking *kaiB* are arrhythmic (3), suggesting that KaiB-HA fusion is at least partially functional.

Frankenbodies were expressed from a promoter that contained a theophylline sensitive riboswitch (26), which allows us to control and tune the expression of the frankenbodies *in vivo* through the addition of theophylline to the media (Fig. 1B). To ensure that the expression of the frankenbodies was not disrupting circadian function, we monitored rhythms of gene expression in strains where frankenbody expression was induced with theophylline. We observed that the expression of either αFLAG or aHA frankenbodies did not result in the alteration of circadian rhythms of gene expression, displaying a WT period (Fig. S1E and F). We additionally performed immunoblot assays using antibodies against FLAG and confirmed that KaiB-FLAG was expressed *in vivo* without the presence of obvious truncation products (Fig. S1G). We, therefore, proceeded with KaiB-FLAG and αFLAG frankenbodies to monitor KaiB localization.

### KaiB localizes to the poles of cells at night

Once we confirmed the functionality of KaiB-FLAG, the luciferase reporter in NS3 was replaced with a construct expressing the αFLAG frankenbody from the theophylline inducible riboswitch (Fig. 1B). After entrainment to a light dark (LD) cycle, where cells were grown in 12 h of light followed by 12 h of dark, KaiB subcellular localization was

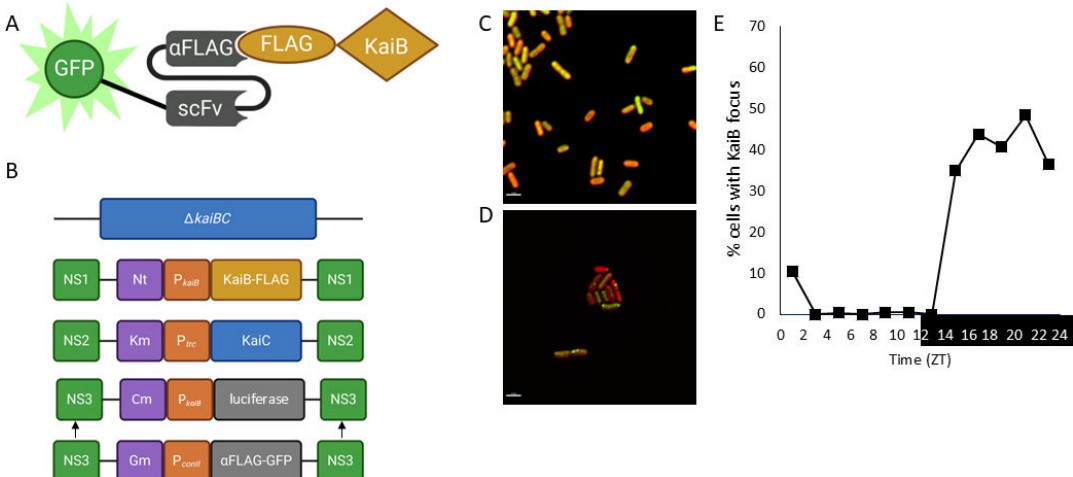

FIG 1 KaiB forms polar foci at night. (A) Schematic of how frankenbodies can be used to localize KaiB. Frankenbodies against the FLAG epitope fused to GFP are stabilized by single-chain variable fragments so that they can be expressed in live cells and used to track the subcellular localization of KaiB. KaiB contains a FLAG epitope tag on its C-terminus. (B) Schematic of how strains were constructed to observe KaiB localization using frankenbodies against the FLAG epitope. Strain AMC705, containing an unmarked in-frame deletion of the *kaiBC* operon (25), was transformed with pLA0067, expressing KaiB-FLAG from its native promoter in NS1, and pAM2595, expressing *kaiC* from the $P_{trc}$ promoter in NS2. In NS3, pAM4663, expressing $P_{kaiB}$-*luc* in NS3, was used to determine the functionality of FLAG tagged *kaiB*. After which pLA0070, expressing $P_{conII}$-RiboB-αFLAG-GFP-frankenbody from NS3 was transformed and replaced pAM4663 to monitor KaiB subcellular localization. (C and D) Representative fluorescent micrographs showing KaiB subcellular localization (green) at (C) ZT 9, during the day, and (D) ZT 21, during the night. Autofluorescence is shown in red. Scale bar = 2.5 mm. (E) KaiB subcellular localization was tracked every 2 h over 24 h in either light (ZT 0–12) or dark (ZT 12–24). Zeitgeber time (ZT) refers to the time relative to when the lights come on. The fraction of the population that showed at least one KaiB focus is plotted, demonstrating that KaiB is primarily diffused during the day but becomes localized as a focus at or near the poles of cells at night.

determined over a 24-h time period under the same diurnal conditions. We observed that, similar to what we observed for KaiA and KaiC (20), KaiB is found diffuse throughout the cell during the day and then formed a focus at or near the poles of cells at night (Fig. 1C through E). KaiB polar localization increased throughout the night plateauing around Zeitgeber Time (ZT) 17 or 5 h after the onset of darkness (Fig. 1E). The expression of the a-FLAG-GFP frankenbody, without KaiB-FLAG, did not support the formation of foci at any time (Fig. S1H), demonstrating that the localization pattern observed is specific to KaiB. These data suggest that KaiB is part of the nighttime clock complex that forms at or near the poles of cells at night and corroborates the vast amount of biochemical data suggesting that KaiB is a critical component of the nighttime clock complex.

## Rbp2 is localized to the poles of cells at night

Rbp2 is part of a family of eukaryotic-like RNAbinding proteins, and Rbp2 was recently identified to associate with KaiC in a localized state (22). *rbp2* mutants exhibit long-period circadian rhythms of gene expression and a 50% decrease in KaiC polar localization at night (22). Substitutions of the amino acids required for binding to RNA in Rbp2 result in a long-circadian period, suggesting RNA binding is required for Rbp2 to function in the clock (22). Determining the localization patterns of Rbp2 alongside KaiC will elucidate the potential role of Rbp2 in KaiC localization.

We generated N-terminal and C-terminal fusions of Yellow Fluorescent Protein (YFP) to Rbp2. These fusions were examined by immunoblot to confirm that the fusion was expressed as full-length fusion and for functionality through the ability to complement a *rbp2* null strain. We observed that while the N-terminally tagged fusion, YFP-Rbp2, was expressed as a full-length fusion without the appearance of truncation products, the C-terminally tagged fusion, Rbp2-YFP, was truncated, suggesting the full-length fusion protein is unstable *in vivo* (Fig. 2A). The expression of YFP-Rbp2 also complemented the null strain, behaving more like the WT strain (Fig. 2B and C), whereas the expression of Rbp2-YFP resulted in long-period rhythms of gene expression, behaving more like a *rbp2*

null strain (Fig. S2). Therefore, we used the N-terminal YFP-Rbp2 fusion to monitor Rbp2 subcellular localization. Similar to what we observe for the Kai oscillator proteins, Rbp2 is diffused throughout the cell during the day and localizes to the cell pole at night (Fig. 4D through F). The polar localization of Rbp2 increased steadily throughout the nighttime portion of the day.

## KaiB and Rbp2 co-localize with KaiC

Using a functional Cyan Fluorescent Protein (CFP) fusion to KaiC (20), we sought to determine if KaiC co-localizes with KaiB, tracked with a GFP fused aFLAG frankenbody, and Rbp2, monitored with the YFP-Rbp2 fusion. Indeed, we observed that KaiC co-localizes with both KaiB (Fig. 3A) and Rbp2 (Fig. 3B) at night, suggesting KaiB and Rbp2 are part of the nighttime clock complex that forms with the KaiA, KaiC, and CikA. Additionally, we quantified the number of foci for KaiC, KaiB, and Rbp2 at ZT 21 and found that when compared directly, KaiC, KaiB, and Rbp2 all form foci in approximately 80% of cells (Fig. 3C).

Interestingly, for both KaiB and Rbp2, we observed a subset of cells that contained more than one focus per cell. While more than one KaiC focus has been reported (20), we appear to be observing a larger fraction of cells with more than one focus. This could be due to several factors including differences in fluorescent microscopes used to track localization as well as the methods used. Here, we are imaging only live cells, and in our previous reports, we used both fixed and live cells. We compared the number of cells that contained more than one focus at ZT 21 for KaiC, KaiB, and Rbp2. We observed that the

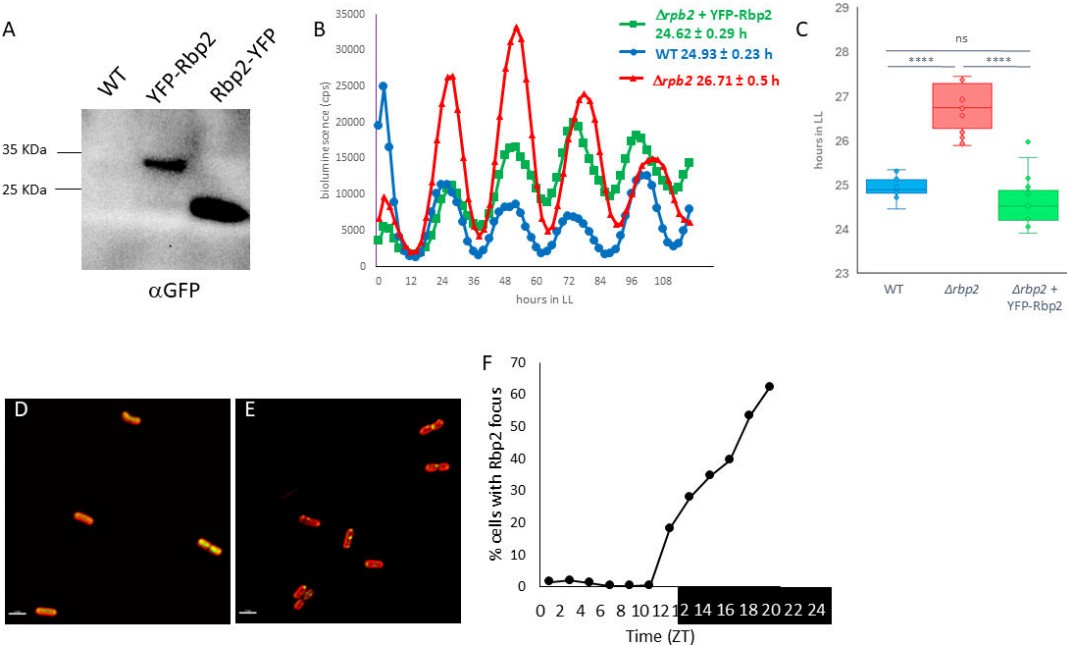

**FIG 2** Rbp2 forms polar foci at night. (A) Immunoblot using an antibody against GFP shows that the N-terminal YFP-Rbp2 fusion is expressed as a full-length fusion protein (37 KDa predicted size) but that the C-terminal Rbp2-YFP fusion is truncated. The WT strain is AMC2036 which does not express YFP. (B) Bioluminescence monitoring of strain carrying a P$_{kaiB}$-luc reporter. The expression of YFP-Rbp2 from NS1 in a D*rbp2* background (green) has a period of 24.62 ± 0.29 h. This is similar to the WT strain, AMC2036 (20), (blue) which has a period of 24.93 ± 0.23 h. The D*rbp2* mutant strain (red) has a period of 26.71 ± 0.5 h suggesting that the YFP-Rbp2 fusion is functional. (C) Bar graph showing statistical analysis of the periods reported in (B). Error bars = standard deviation. ****$P < 0.00002$, One-way ANOVA with Tukey's post-hoc analysis. ns, not significant. (D and E) Representative fluorescent micrographs showing Rbp2 subcellular localization (green) at (D) ZT 9, during the day, and (E) ZT 21, during the night. Autofluorescence is shown in red. Scale bar = 2.5 mm. (F) Rbp2 subcellular localization was tracked every 2 h over 24 h in either light (ZT 0–12) or dark (ZT 12–24). Zeitgeber time (ZT) refers to the time relative to when the lights come on. The fraction of the population that showed at least one Rbp2 focus is plotted, demonstrating that Rbp2 is primarily diffused during the day but becomes localized as a focus at night.

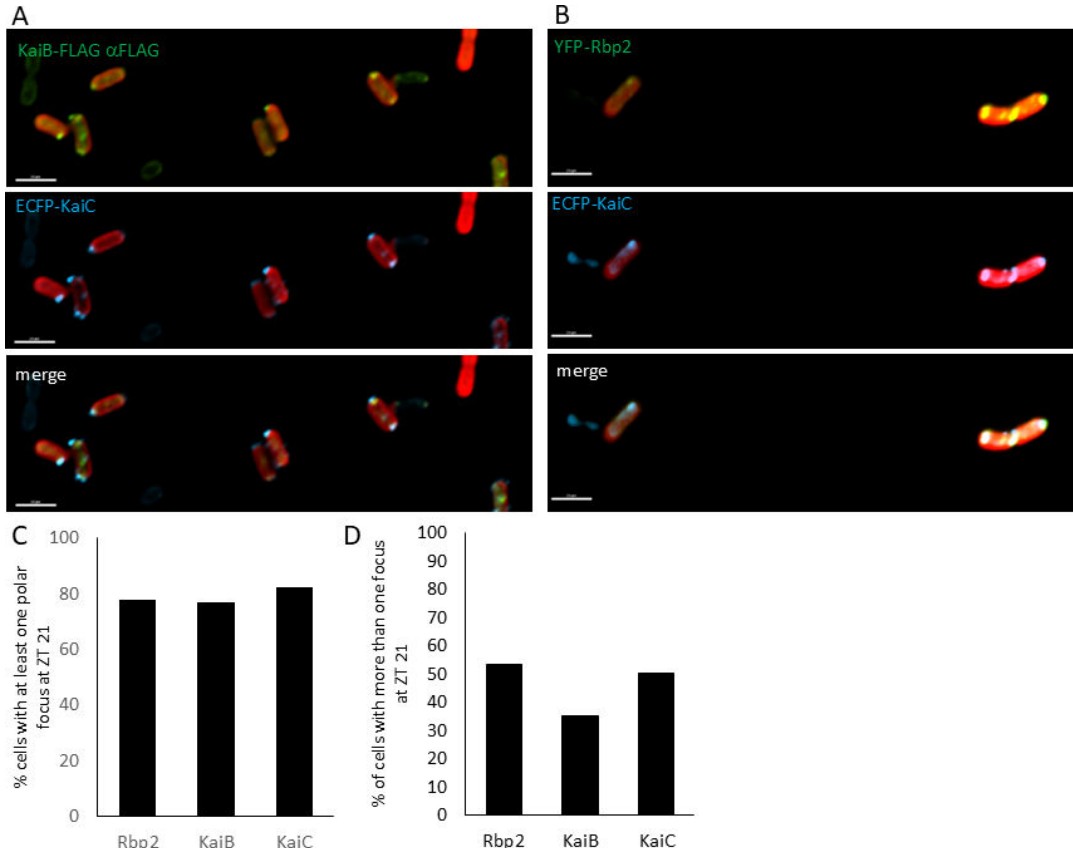

**FIG 3** KaiC colocalizes with KaiB and Rbp2 at night. (A and B) Representative fluorescent micrographs showing that (A) KaiB (green) or (B) YFP-Rbp2 (green) co-localizes with KaiC-ECFP (blue). Autofluorescence is shown in red. Scale bar = 2.5 mm. (C) The percentage of cells that contain at least one focus for Rbp2, KaiB, and KaiC is plotted at ZT21. No statistically significant difference was observed (one way ANOVA with Tukey's post-hoc analysis). (D) The percentage of cells that contain more than one foci at ZT21 is plotted for Rbp2, KaiB, and KaiC. No statistically significant difference was observed (one-way ANOVA with Tukey's post-hoc analysis).

number of cells that contained more than one focus per cell was 50% for KaiC and Rbp2 and 30% for KaiC (Fig. 3D). These differences did not appear to be statistically significant.

## Substitution of Rbp2's RNAbinding domain results in decreased Rbp2 localization

We have previously reported that the RNAbinding activity of Rbp2 is required for Rbp2 to execute its function in the circadian clock as the expression of RNA binding mutant variants does not complement the defects in circadian rhythms of gene expression observed in a *rbp2* null strain (22). We sought to determine if Rbp2's RNA binding activity was required for Rbp2's localization at night. We observed that the expression of an Rbp2 RNA binding mutant variant, $Rbp2^{R42AF44AF46A}$, results in decreased polar localization at ZT21 (Fig. 4). Specifically, at this timepoint, we observe that 58.5% of cells contain a focus for wild-type YFP-Rbp2 (Fig. 4A and C) and that only 14% of cells have a focus for cells expressing $YFP-Rbp2^{R42AF44AF46A}$ (Fig. 4B and C), where primarily diffuse localization is observed. While the RNA binding activity of Rbp2 is not essential for localization, as localization is still observed in some cells, these data suggest that the RNA binding activity of Rbp2 is required for robust complex formation at night and implicates RNA in the localization mechanism.

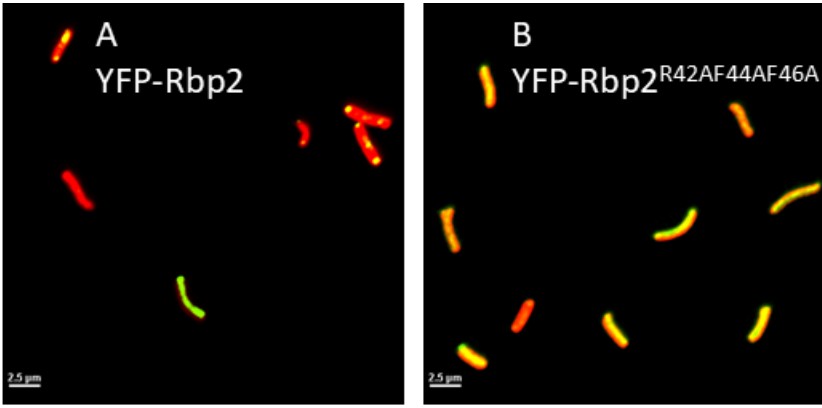

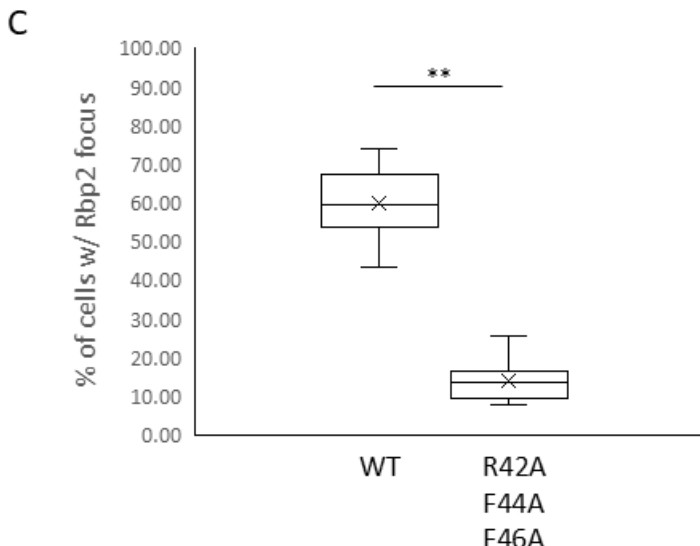

**FIG 4** RNA binding activity of Rbp2 promotes Rbp2's polar localization. (A and B) Representative fluorescent micrographs of cells expressing (A) YFP-Rbp2 or (B) YFP-Rbp2$^{R42AF44AF46A}$ at ZT21. Autofluorescence is shown in red. Scale bar = 2.5µm. (C) The percentage of cells that contain at least one Rbp2 focus at ZT21 is plotted, demonstrating that RNA binding activity of Rbp2 is required to observe robust polar localization of Rbp2. Error bars = standard deviation. **$P < 0.01$ (one-way ANOVA with Tukey's post-hoc analysis).

## DISCUSSION

Here, we determined the subcellular localization patterns of KaiB and Rbp2 throughout the day-night cycle using fluorescence microscopy and image analysis. We observe that both proteins exhibit dynamic subcellular localization patterns, localizing to the cytoplasm during the day and forming a polar focus, or series of foci, at night. These localization patterns mirror those in previous studies examining the localization of KaiA, KaiC, and CikA (20, 21). We observe that KaiC co-localizes with both KaiB and Rbp2, suggesting that these proteins are all part of the same nighttime polar protein complex.

A high protein copy number of KaiC and KaiB is required to maintain rhythmicity in *S. elongatus* (27). However, KaiB and KaiC protein abundance peaks in the early night and is decreasing when polar localization is at its peak (27). Localization of the KaiABC proteins to the pole could provide a method for counteracting lower copy numbers to maximize protein-protein interaction in the late night. The novel discovery that KaiB, KaiC, and Rbp2 form multiple foci in a subset of cells may provide insight into the mechanisms behind the subcellular localization of circadian proteins. It may indicate that protein

complexes form throughout the cell and then migrate to the pole rather than migrating to the pole and then associating.

Strains lacking *rbp2* exhibit a long circadian period of gene expression and decreased KaiC localization at night, suggesting that Rbp2 could be a driver of circadian protein localization (22). The co-localization of Rbp2 with KaiC and the fact that KaiC localization is impaired in strains lacking *rpb2* (22) provides further support that Rbp2 binds to the KaiC complex and guides or tethers it to the pole of the cell. Since Rbp2 is primarily diffuse throughout the cell during the day, Rbp2 might act as a guide rather than an anchor permanently localized to the pole. However, it is also possible that KaiC may be responsible for Rbp2 subcellular localization where both are required for robust oscillations in subcellular localization to be observed. Substitution of the RNA recognition motif of Rbp2 causes similar circadian disruption to what is observed in a *rbp2* deletion strain (22). This suggests the ability of Rbp2 to bind to RNA is central to its circadian function. Similarly, we observed that substitution of the amino acids in the RNA recognition motif of Rbp2, Rbp2$^{R42A-F44A-F46A}$, resulted in decreased Rbp2 polar localization at night, suggesting that RNA binding by Rbp2 is required to mediate polar localization of the Rbp2 and potentially affects circadian function by disrupting localization patterns.

Various prokaryotic RNAs have been shown to be under spatiotemporal control. For example, *psbA* mRNA, which encodes a core photosystem subunit, exhibits a similar pattern of subcellular localization as that we described for KaiB, KaiC, and Rbp2 (28). Taken together with our results, we propose a model by which Rbp2 in complex with the KaiABC oscillator binds to an RNA which localizes, along with the protein complex, to the pole of the cell (Fig. 5). In this model, the RNA is used as a guide or scaffold to help the proteins or protein complexes localize as a focus at or near the poles of cells.

The use of endogenously produced frankenbodies for live cell imaging introduces a method for *in vivo* observation of proteins like KaiB which cannot function properly fused to a fluorophore. This method could also be used to monitor transient proteins previously unable to be imaged due to the relatively long folding process of fluorophores like GFP. Frankenbodies were originally designed for use in eukaryotic cell lines and whole organisms (24). Here, we demonstrate that they can be adapted for use in bacteria. Endogenous antibody expression is a powerful, flexible technique with the potential to enhance the bacterial imaging toolkit and impact multiple fields of research.

## MATERIALS AND METHODS

### Bacterial strains, growth conditions, and DNA manipulations

*Synechococcus elongatus* PCC 7942 was obtained from the Golden Lab at UCSD and grown in BG-11 media supplemented with appropriate antibiotics at 30°C under 50 to 300 µE of light for 3–5 days (29). Strains were constructed by expressing genes in one of three *S. elongatus* neutral sites (NS) NS1, NS2, or NS3. Antibiotics used in this study include chloramphenicol (Cm), kanamycin (Km), spectinomycin and streptomycin (Sp and Sm), gentamycin (Gm), and nourseothricin (Nt). Concentrations were used as previously described (30). Plasmids and *S. elongatus* strains are described in Table S1. Plasmids were designed using the CYANO-VECTOR assembly portal (http://golden.ucsd.edu/CyanoVECTOR/), constructed with the GeneArt Seamless Cloning and Assembly Kit (Life Technologies) and propagated in *E. coli* XL1 Blue cells as previously described (30). All knock-out strains were checked for complete segregation of mutant loci by PCR.

### Circadian bioluminescence monitoring

Bioluminescence was monitored using a P$_{kaiB}$-luciferase fusion reporter under constant light and temperature (30°C). Cultures were entrained to 2–3 cycles of 12 h light and 12 h dark at 30°C to synchronize the population as previously described (31). Bioluminescence was monitored every 2 h from a TECAN Spark bioluminescence plate reader for 5–7 days.

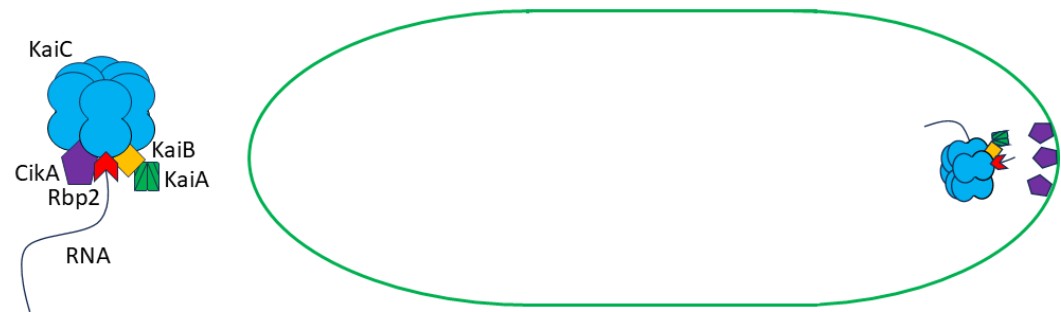

**FIG 5**  Model for how Rbp2 and the Kai oscillator localize to the poles of cells at night. Model proposing that RNA bound to Rbp2 may be used as a scaffold to allow Rbp2 bound to the oscillator to localize as a focus at or near the poles of cells. While KaiA, KaiB, KaiC, and Rbp2's localization patterns change throughout the day, CikA has been shown to remain at the poles throughout the day-night cycle (20), where it is joined by Rbp2 and the KaiC complex as part of the nighttime clock complex.

Data were plotted in Excel and analyzed for rhythmicity using a MFourFit algorithm in BioDare2 (https://biodare2.ed.ac.uk) (32)

## Immunoblotting

Whole-cell extract preparation and immunoblot analysis were performed as previously described (33). Equal amounts of total protein (10 µg) were separated by SDS-PAGE (10%), transferred to a polyvinyldifluoride (PVDF) membrane, and blocked overnight at 4°C in 5% (wt/vol) non-fat dry milk in TBS-T (for a-GFP) or PBS-T (for a-FLAG). Membranes were washed with TBS-T (for a-GFP) or PBS-T (for a-FLAG) and probed with either 1:2,000 rabbit a-GFP (Invitrogen) in 2.5% non-fat milk in TBS-T or 1:10,000 mouse a-FLAG (Rockland) in 1% non-fat milk in PBS-T at 25°C for 1 h, followed by several washes in either TBS-T or PBS-T and probed with horseradish peroxidase (HRP)-conjugated secondaries, 1:2,000 a-Rabbit (Invitrogen) in 2.5% non-fat milk in TBS-T or 1:10,000 goat a-Mouse (Thermo Fisher Scientific) in 1% non-fat milk in PBS-T for 1 h. Chemiluminescent detection was performed using Pierce Super Signal West Atto detection reagents (Thermo Scientific).

## Fluorescence microscopy and image analysis

Cells were either placed on a pad of 1.2% agarose in BG-11 media or 1.5 µL of culture was placed onto a glass slide and covered with a coverslip. Microscopy was performed with a customized Nikon TiE inverted wide-field microscope with a Near-IR-based Perfect Focus system. Images were acquired with a BSI 95% Quantum sCMOS camera (Photometrics) using a Nikon CF160 Plan Apochromat Lambda 100 × oil immersion objective (1.45 N.A.). Autofluorescence of thylakoid membranes was imaged using a 554 nm LED light source (SpectraX) for excitation and a standard TRITC emission filter (609/54 nm). YFP localization was imaged using a 500 nm LED light source (SpectraX) for excitation and a standard YFP emission filter (535/30 nm). CFP localization was imaged using a 436 nm LED light source (Spectra X) for excitation and a standard CFP emission filter (480/40 nm). Exposure times were limited to avoid bleed-through from thylakoid fluorescence (34). For time-course experiments, samples were entrained to opposite light dark cycles and sampled every 2 h in either light (ZT 0-12) or dark (ZT 12-24). Two hundred to six hundred cells were manually counted, from biological duplicates or triplicates, at each timepoint to determine the percentage of the population that contained at least one polar focus. For imaging KaiB localization, strains expressing KaiB-FLAG and frankenbody controls were induced 12–24 h prior to imaging with 2 mM theophylline resuspended in DMSO. Images were colorized in Nikon Elements software and prepared for figure assembly.

## ACKNOWLEDGMENTS

We thank Tim Stasevich at Colorado State University, Fort Collins for the gift of the frankenbody expressing plasmid and members of the Cohen lab for their thoughtful comments on the manuscript.

This work is supported by NSF CAREER Award MCB-1845953 to S.E.C.

## AUTHOR AFFILIATIONS

[1]Department of Biological Sciences, California State University, Los Angeles, Los Angeles, California, USA

[2]Center for Circadian Biology, University of California, San Diego, La Jolla, California, USA

## AUTHOR ORCIDs

Susan E. Cohen http://orcid.org/0000-0003-1783-0011

## FUNDING

| Funder | Grant(s) | Author(s) |
| --- | --- | --- |
| National Science Foundation | MCB-1845953 | Susan E. Cohen |

## AUTHOR CONTRIBUTIONS

Harry J. Bevir, Conceptualization, Data curation, Formal analysis, Supervision, Writing – original draft, Writing – review and editing | Christopher C. Hooper, Data curation, Formal analysis | Parker Saikley, Data curation, Formal analysis | Tanya Chaljian, Data curation, Resources | Yahan Lin, Data curation, Writing – review and editing | Susan E. Cohen, Conceptualization, Formal analysis, Funding acquisition, Supervision, Writing – original draft, Writing – review and editing

## ADDITIONAL FILES

The following material is available online.

### Supplemental Material

**Fig. S1 (Spectrum01845-25-s0001.tif).** Validation of strains used to determine KaiB subcellular localization.
**Fig. S2 (Spectrum01845-25-s0002.tif).** Validation of strains used to determine Rbp2 subcellular localization.
**Supplemental material (Spectrum01845-25-s0003.docx).** Supplemental figure legends.

### Open Peer Review

**PEER REVIEW HISTORY (review-history.pdf).** An accounting of the reviewer comments and feedback.

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
