## [Reviewer comments · Microbiology Spectrum]

Microbiology Spectrum

Circadian clock proteins KaiB and Rbp2 of *Synechococcus elongatus* display oscillations in their subcellular localization patterns

Harry Bevir, Christopher Hooper, Parker Saikley, Tanya Chaljian, Yahan Lin, and Susan Cohen

Corresponding Author(s): Susan Cohen, Cal State LA

Review Timeline:

Submission Date:	June 16, 2025
Editorial Decision:	July 12, 2025
Revision Received:	December 3, 2025
Accepted:	December 8, 2025

Editor: Ilana Kolodkin-Gal

Reviewer(s): The reviewers have opted to remain anonymous.

Transaction Report:

DOI: <https://doi.org/10.1128/spectrum.01845-25>

Re: Spectrum01845-25 (**Circadian clock proteins KaiB and Rbp2 of *Synechococcus elongatus* display oscillations in their subcellular localization patterns**)

Dear Dr. Susan Cohen:

Thank you for the privilege of reviewing your work. Both expert reviewers were positive and found the work novel and of interest to the field. Below you will find my comments, instructions from the Spectrum editorial office, and the reviewer comments.

Revision Guidelines

Sincerely,
Ilana Kolodkin-Gal
Editor
Microbiology Spectrum

Reviewer #1 (Comments for the Author):

The cyanobacterial circadian clock is composed of clock proteins, whose timekeeping function is carried out in the cytoplasm. Recently, some of these clock proteins, namely KaiA, KaiC, and CikA, were found to localize to the cell poles during the nocturnal phase of the diurnal cycle. The role of this polar localization is not well understood, but it is of interest in bacterial research because it is also observed in heterologous expression systems in *E. coli* cells. In this study, the authors showed that

the clock protein KaiB and the clock-acting factor Rbp2 coexist with KaiC and exhibit polar localization as described above; KaiB was successfully labeled with GFP for the first time by introducing a linker protein into the model organism cyanobacteria that does not disrupt the steric structure of the complex. This study represents a unique approach from the viewpoint of clock protein localization to the enigma of spatio-temporal dynamism in the photosynthetic bacterial cells under diurnal conditions. These results are interesting. However, I have some concerns regarding the organization of the manuscript and the experimental methods.

The manuscript is well written. However, the sentences are long due to explanatory text inserted into the text in places, which takes time to comprehend. Simple and concise sentence structure leads to readability. This is especially the case in Figure Legends. Example; page 16 middle; B) Bioluminescence monitoring of strain carrying a PkaiB-luc reporter (AMC,,,,,,).

Transparency of reporting: Statistics are missing in majority of experiments and they should be added.

Major concerns

The content is divided in that there is no direct link between KaiB and Rbp2. Although the fluorescent labelling method for KaiB is novel and important in that it is the first time KaiB has been near-native, it is not directly related to the content. Therefore, presenting this as a standalone figure would distract the reader from the main story and make the transition to Rbp2 time-consuming. To make it easier to read, Figures 1-2 should either be moved to the supplementary material or combined into Figure 1-3.

Figure 7 should depict potential targets, such as genome-compactation and cytoplasmic clock protein complexes. These should be labeled with a question mark.

As a negative control, it should be demonstrated that polar localization does not occur with the expression of anti-FLAG-GFP alone.

Minor concern

Add a scale to the horizontal axis of the graph.

page 16 middle; B) Bioluminescence monitoring of strain carrying a PkaiB-luc reporter (AMC,,,,,,). This sentence is too long.

Fig, 2

Pannels A and B overlap.

(C) Western blotting. ~10 kDa protein marker position is missed. The band of KaiB-FLAG should be indicated by an arrow.

Typo;

Page 15 bottom, 2mM needs a space.

page 18, Supplemental Figure Legends; B9 frankenbody (orange? Blue?). WT strain in blue(Orange?). In addition, Anti HA is in dark brown(?).

Reviewer #2 (Comments for the Author):

The manuscript by Bevir et al. investigates the subcellular localization dynamics of the circadian clock proteins KaiB and Rbp2, revealing oscillations in their localization patterns. While previous studies examined the localization of core oscillator proteins KaiA and KaiC, visualization of KaiB had been hindered by its functional impairment when fused to fluorescent proteins. To overcome this limitation, the authors employed genetically encoded probes (frankenbodies), successfully demonstrating that KaiB forms polar foci and colocalizes with KaiC at night.

Furthermore, using a YFP-Rbp2 fusion, the authors show that Rbp2 also forms polar foci and colocalizes with KaiC. Additionally, they demonstrate that the RNA-binding motif of Rbp2 is required for robust foci formation. Together, these findings provide novel insights into the dynamics of circadian clock components and deepen our understanding of clock mechanisms in cyanobacteria.

Major comments

1. Figure 7 illustrates a proposed KaiABC-Rbp2 complex tethered by RNA to a polar-localized CikA. While the data support the existence of a KaiABC-Rbp2 complex and demonstrate the requirement of the RNA-binding motif for Rbp2 localization, direct interactions between CikA and RNA were not examined. In its current form, the figure may therefore be misleading. I recommend revising Figure 7 to more accurately reflect the experimental evidence.

2. I did not find statistical analyses comparing differences in rhythms, for example, whether $\Delta rbp2$ (26.71 {plus minus} 0.5 h) significantly differs from $\Delta rbp2 + YFP-Rbp2$ (24.62 {plus minus} 0.29 h). Please include appropriate statistical tests to support

these comparisons. Additionally, bar graphs in Figures 5C, 5D, and 6C lack error bars. While it is mentioned that 200-600 cells were counted, it is unclear how many biological replicates these data represent. Please clarify the number of biological replicates and include appropriate statistical analyses and error bars where applicable.

Minor comments

1. The Introduction could be shortened. Many mechanistic details, while interesting, are not essential for understanding the current study.
2. On page 7, the text describes a luciferase reporter in NS3 and refers the reader to Figure 1B, which depicts α FLAG-GFP in NS3. Only later is it clarified that the luciferase reporter was replaced with an α FLAG frankenbody. To improve clarity, I suggest revising Figure 1B to show both insertion variants in NS3 or moving one variant to a supplementary figure.
3. The sentence bridging page 9 and page 10 states: "Even in cells with multiple foci, the foci are primarily at or near the pole or otherwise associated with the thylakoid membrane." I did not find evidence supporting the association with the thylakoid membrane. Please clarify or remove the statement "or otherwise associated with the thylakoid membrane."
4. Page 11: "The co-localization of Rbp2 with KaiC provides further support that Rbp2 binds to the KaiC complex and guides or tethers it to the pole of the cell." Is there direct evidence supporting this sequence of events? Could it be that KaiC guides Rbp2 to the pole instead? Please clarify or consider revising this statement to avoid overinterpretation.
5. Page 8: "KaiB polar localization increased throughout the night plateauing around Zeitgeber Time (ZT) 17 or 5 hours after the onset of darkness." Please add a reference to Figure 3C at the end of this sentence.
6. Page 12: The reader should be referred to Fig. 7 (not Fig.6).
7. Several sentences in the Results and Discussion sections would benefit from careful revision to improve clarity and readability. Below are a few examples; however, these are not exhaustive. I recommend that the authors review and edit the entire Results and Discussion sections to ensure clarity and ease of reading throughout.
 - a) The last sentence at the bottom of page 6 - consider splitting this sentence or rephrasing.
 - b) Top of page 8: "After entrainment to a light dark (LD) cycle, where cells were grown in 12 hours of light followed by 12 hours of dark, and KaiB subcellular localization was determined over a 24-hour time period under the same diurnal conditions." I think the word "and" before "KaiB" should be removed.
 - c) Page 8: "Rbp2 is part of a family of eukaryotic-like RNA binding proteins that was recently identified to associate with KaiC in a localized state (23)." Rephrase - in the current sentence, "that" refers to the family of eukaryotic-like RNA binding proteins and not, as intended, to Rbp2.
 - d) Page 8: "Mutations of the amino acids..." Consider using "substitutions" or "replacements" instead of "mutations" for greater accuracy.

CAL STATE LA
CALIFORNIA STATE UNIVERSITY, LOS ANGELES

5151 State University Drive, Los Angeles, CA 90032-8502 (323) 343-2091 www.calstatela.edu

Susan E. Cohen, Ph.D., Associate Professor
Department of Biological Sciences
scohen8@calstatela.edu
Office: (323) 343-2091

November 24, 2025

Dr. Ilana Kolodkin-Gal
Editor, Microbiology Spectrum

Dear Dr. Kolodkin-Gal,

We are submitting a revised version of our manuscript entitled, “Circadian clock proteins KaiB and Rbp2 of *Synechococcus elongatus* display oscillations in their subcellular localization patterns” by Harry J. Bevir, Christopher C. Hooper, Parker Saikley, Tanya Chaljian, Yahan Lin and Susan E. Cohen. Please convey our thanks to the reviewers for their enthusiasm for the work and for their thoughtful comments, which have helped us to improve the manuscript. We have addressed the comments from the reviewers. Specifically, we have reorganized the presentation of the data and included critical controls and statistical analysis and have included a new co-author Yahan Lin. I hope that you will now find the manuscript suitable for publication in *Microbiology Spectrum*.

Sincerely,

Susan Cohen

Detailed responses to reviewer’s comments:

Reviewer 1:

1. *“The manuscript is well written. However, the sentences are long due to explanatory text inserted into the text in places, which takes time to comprehend. Simple and concise sentence structure leads to readability. This is especially the case in Figure Legends.*

Example; page 16 middle; B) Bioluminescence monitoring of strain carrying a PkaiB-luc reporter (AMC,,,,,.”

Thank you for this comment. We have reviewed and revised the text, in particular the figure legends to ensure their simplicity to improve readability.

2. *“Transparency of reporting: Statistics are missing in majority of experiments and they should be added.”*

We apologize for this. Statistics have now been included in Figures 2-4 as well as supplementary figures 1 and 2.

3. *“The content is divided in that there is no direct link between KaiB and Rbp2. Although the fluorescent labelling method for KaiB is novel and important in that it is the first time KaiB has been near-native, it is not directly related to the content. Therefore, presenting this as a standalone figure would distract the reader from the main story and make the transition to Rbp2 time-consuming. To make it easier to read, Figures 1-2 should either be moved to the supplementary material or combined into Figure 1-3.”*

Thank you for understanding the importance of demonstrating KaiB localization. We condensed the data showing KaiB subcellular localization into a revised Figure 1 and moved the remaining figures validating the frankenbody strains to Supplementary Figure 1.

4. *“Figure 7 should depict potential targets, such as genome-compaction and cytoplasmic clock protein complexes. These should be labeled with a question mark.”*

In our previous publications, we have ruled out nucleoid occlusion as well as many other cellular targets as the mechanism that drives the localization of the clock complexes and thus we are concerned about showing them in this model figure.

5. *“As a negative control, it should be demonstrated that polar localization does not occur with the expression of anti-FLAG-GFP alone.”*

We have now included this data in Supplementary Figure 1H. The text now reads “Expression of the α -FLAG-GFP frankenbody, without KaiB-FLAG, did not support the formation foci at any time (Supplementary Fig. 1H), demonstrating that the localization pattern observed is specific to KaiB.”

6. “page 16 middle; B) Bioluminescence monitoring of strain carrying a *P_{kaiB}-luc* reporter (AMC,,,,,. This sentence is too long.”

We have revised the sentence to improve clarity. The text now reads “B) Bioluminescence monitoring of strain carrying a *P_{kaiB}-luc* reporter. Expression of YFP-Rbp2 from NS1 in a $\Delta rbp2$ background (green) has a period of 24.62 ± 0.29 hours. This is similar to the WT strain, AMC2036 (20), (blue) which has a period of 24.93 ± 0.23 hours. The $\Delta rbp2$ mutant strain (red) has a period of 26.71 ± 0.5 hours suggesting that the YFP-Rbp2 fusion is functional.”

7. “Fig. 2 Pannels A and B overlap.”

Our apologies, the figures have been adjusted so that they no longer overlap.

8. “Western blotting. ~10 kDa protein marker position is missed. The band of KaiB-FLAG should be indicated by an arrow.”

The 10KDa ladder position has been added as well as an arrow indicating the position of KaiB-FLAG.

9. “Typo. Page 15 bottom, 2mM needs a space.”

Thank you for catching this. It has been corrected. The text now reads “...strains expressing KaiB-FLAG and frankenbody controls were induced 12-24 hours prior to imaging with 2 mM theophylline resuspended in DMSO.”

10. “page 18, Supplemental Figure Legends; B9 frankenbody (orange? Blue?). WT strain in blue(Orange?). In addition, Anti HA is in dark brown(?)”

All of the figure legends have all be revised for clarity. For this particular revised figure legend it now reads “A) Bioluminescence monitoring of strain carrying a *P_{kaiB}-luc* reporter shows that expression of Rbp2-YFP in a $\Delta rbp2$ mutant background (purple) has a period of 25.63 ± 0.54 hr, which is more similar to the $\Delta rbp2$ mutant strain (red), 26.71 ± 0.5 hr, suggesting that the C-terminal fusion is not functional. WT strain is AMC2036 (blue), has a period of 24.93 ± 0.23 hr.”

Reviewer 2:

1. “Figure 7 illustrates a proposed KaiABC-Rbp2 complex tethered by RNA to a polar-localized CikA. While the data support the existence of a KaiABC-Rbp2 complex and demonstrate the requirement of the RNA-binding motif for Rbp2 localization, direct

interactions between CikA and RNA were not examined. In its current form, the figure may therefore be misleading. I recommend revising Figure 7 to more accurately reflect the experimental evidence.”

Thank you for this comment. Our previous work has shown that CikA also co-localizes with the clock complex at night (although it is also localized throughout the day), which is why we included it, but it is correct that there is no evidence of CikA-RNA associations. We have revised the figure to not suggest that CikA is associating with RNA.

2. *“I did not find statistical analyses comparing differences in rhythms, for example, whether $\Delta rbp2$ (26.71 {plus minus} 0.5 h) significantly differs from $\Delta rbp2 + YFP-Rbp2$ (24.62 {plus minus} 0.29 h). Please include appropriate statistical tests to support these comparisons. Additionally, bar graphs in Figures 5C, 5D, and 6C lack error bars. While it is mentioned that 200-600 cells were counted, it is unclear how many biological replicates these data represent. Please clarify the number of biological replicates and include appropriate statistical analyses and error bars where applicable.”*

We apologize for the confusion. Statistical analysis have now been included in Figures 2-4 as well as supplementary figures 1 and 2. We have also updated the materials and methods to include how many replicates were imaged.

3. *“The Introduction could be shortened. Many mechanistic details, while interesting, are not essential for understanding the current study.”*

We have shortened the introduction accordingly.

4. *“On page 7, the text describes a luciferase reporter in NS3 and refers the reader to Figure 1B, which depicts α FLAG-GFP in NS3. Only later is it clarified that the luciferase reporter was replaced with an α FLAG frankenbody. To improve clarity, I suggest revising Figure 1B to show both insertion variants in NS3 or moving one variant to a supplementary figure.”*

Thank you for the suggestion, in Figure 1B, we now show that the α -FLAG-GFP construct replaces the luciferase reporter used to track circadian rhythms of gene expression.

5. *“The sentence bridging page 9 and page 10 states: “Even in cells with multiple foci, the foci are primarily at or near the pole or otherwise associated with the thylakoid membrane.” I did not find evidence supporting the association with the thylakoid*

membrane. Please clarify or remove the statement "or otherwise associated with the thylakoid membrane.""

“or otherwise associated with the thylakoid membrane was” was removed. In our previous work we had quantified this association for KaiC, but since we do not here we removed the statement.

6. *“Page 11: "The co-localization of Rbp2 with KaiC provides further support that Rbp2 binds to the KaiC complex and guides or tethers it to the pole of the cell." Is there direct evidence supporting this sequence of events? Could it be that KaiC guides Rbp2 to the pole instead? Please clarify or consider revising this statement to avoid overinterpretation.”*

There is no evidence for one mechanism over the other. We therefore revised the sentence. The sentence now reads “Since Rbp2 is primarily diffuse throughout the cell during the day, Rbp2 might act as a guide rather than an anchor permanently localized to the pole. However, it is also possible that KaiC may be responsible for Rbp2 subcellular localization where both are required for robust oscillations in subcellular localization to be observed.”

7. *“Page 8: "KaiB polar localization increased throughout the night plateauing around Zeitgeber Time (ZT) 17 or 5 hours after the onset of darkness." Please add a reference to Figure 3C at the end of this sentence.”*

The sentence has been revised. It now reads “KaiB polar localization increased throughout the night plateauing around Zeitgeber Time (ZT) 17 or 5 hours after the onset of darkness (Figure 1E).” The figure’s were rearranged in accordance with comments from reviewer 1.

8. *“Page 12: The reader should be referred to Fig. 7 (not Fig.6).”*

Thank you for this correction. The text now reads “Taken together with our results, we propose a model by which Rbp2 in complex with the KaiABC oscillator binds to an RNA which localizes, along with the protein complex, to the pole of the cell (Figure 5).”

9. *“Several sentences in the Results and Discussion sections would benefit from careful revision to improve clarity and readability. Below are a few examples; however, these are not exhaustive. I recommend that the authors review and edit the entire Results and Discussion sections to ensure clarity and ease of reading throughout.
a) The last sentence at the bottom of page 6 - consider splitting this sentence or*

rephrasing.

b) Top of page 8: "After entrainment to a light dark (LD) cycle, where cells were grown in 12 hours of light followed by 12 hours of dark, and KaiB subcellular localization was determined over a 24-hour time period under the same diurnal conditions." I think the word "and" before "KaiB" should be removed."

Thank you for the suggestion. We have made the suggested edits and have reviewed the text to ensure clarity and readability.

10. *"Page 8: "Rbp2 is part of a family of eukaryotic-like RNA binding proteins that was recently identified to associate with KaiC in a localized state (23)." Rephrase - in the current sentence, "that" refers to the family of eukaryotic-like RNA binding proteins and not, as intended, to Rbp2."*

Thank you, the text now reads, "Rbp2 is part of a family of eukaryotic-like RNA binding proteins and Rbp2 was recently identified to associate with KaiC in a localized state (23)."

11. *"Page 8: "Mutations of the amino acids..." Consider using "substitutions" or "replacements" instead of "mutations" for greater accuracy."*

The suggested change was made. The text now reads "Substitutions of the amino acids required for binding to RNA in Rbp2 result in a long-circadian period, suggesting RNA binding is required for Rbp2 to function in the clock (23)."

Re: Spectrum01845-25R1 (**Circadian clock proteins KaiB and Rbp2 of *Synechococcus elongatus* display oscillations in their subcellular localization patterns**)

Dear Prof. Susan Cohen:

You and your research team conducted additional experiments to strengthen the robustness of their original findings. This included incorporating controls for the expression of the α -FLAG-GFP frankenbody without KaiB-FLAG, adding the appropriate markers to the Western blot, and revising the text and legends to address the concerns raised by Reviewer 1. They also included statistical analyses, clarified the details of strain construction, and removed statements that relied on previous works from the data interpretation to address in full the concerns of reviewer 2.

The manuscript now meets the publication criteria set by ASM. I appreciate your efforts to improve and revise the manuscript and hope you and your team will consider submitting additional work to Spectrum on the molecular mechanisms of the circadian clock in the future.

Your manuscript has been accepted, and I am forwarding it to the ASM production staff for publication. Your paper will first be checked to make sure all elements meet the technical requirements. ASM staff will contact you if anything needs to be revised before copyediting and production can begin. Otherwise, you will be notified when your proofs are ready to be viewed.

Sincerely,
Ilana Kolodkin-Gal
Editor
Microbiology Spectrum